**Subject Category:**
Biology (whole organism)

behaviour/ecology

facilitation, vertical displacement, schooling fish, seabirds, pursuit divers

**Author for correspondence:**
A. M. McInnes
e-mail: amcinnes3@gmail.com

# Up for grabs: prey herding by penguins facilitates shallow foraging by volant seabirds

## A. M. McInnes and P. A. Pistorius

DST/NRF Centre of Excellence at the Percy FitzPatrick Institute, Department of Zoology, Nelson Mandela University, Summerstrand 6031, South Africa

AMM, 0000-0002-9125-9629

Visual and olfactory signals are commonly used by seabirds to locate prey in the horizontal domain, but foraging success depends on prey depth and the seabird's ability to access it. Facilitation by diving seabirds has long been hypothesized as a mechanism to elevate deep prey to regions more accessible to volant seabirds, but this has never been demonstrated empirically. Footage from animal-borne video loggers deployed on African penguins was analysed to establish if volant seabird encounters involved active cuing by seabirds on penguins to obtain prey and, during mutual prey encounters, if interactions were driven by the vertical displacement of prey by penguins. Independent of prey biomass estimates, we found a strong inverse relationship between penguin group size, a proxy for visibility, and the time elapsed from the start of penguins' dive bouts to their first encounter with other seabirds. Most mutual prey encounters (7 of 10) involved schooling prey elevated from depths greater than 33 m by penguins and only pursued by other seabird species once prey was herded into shallow waters. This is likely to enhance foraging efficiency in volant seabird species. As such, penguins may be integral to important processes that influence the structure and integrity of marine communities.

## 1. Introduction

Seabirds use different sensory cues, such as local enhancement (i.e. visual cuing on predator aggregations) and olfaction, to locate foraging localities, but accessing prey at depth is limited by their foraging mode. Diving capabilities of seabirds range from pursuit diving birds, such as penguins and alcids, with maximum dive depths greater than 100 m [1], to surface-feeding birds, such as terns and gulls that can only access prey near the surface [2]. Temperate marine ecosystems are often dominated by a few mid-trophic prey species that undergo dial vertical migrations and are often located at shallower depths at night, descending into deeper waters during the day, as is generally found in upwelling systems

[3]. In such systems, many diurnal volant seabirds have limited access to this prey and are known to associate with diving birds and mammals, which are thought to facilitate prey access by herding prey into shallow waters [2,4]. The implications of these associations have particular relevance at a community level and consequently to the conservation importance of facilitating species [5]. Much of the evidence for these associations is based on boat-based observations, whereas the study of the underlying mechanisms driving these associations has until recently been limited by the lack of *in situ* sub-surface observations.

African penguins *Spheniscus demersus* are endemic to the Benguela upwelling ecosystem where they feed predominantly on small epipelagic fish such as sardine *Sardinops sagax* and anchovy *Engraulis encrasicolus*. They frequently dive to more than 30 m [6] and can herd schools of fish from these depths into shallow waters [7]. Volant seabirds such as gulls and terns are attracted to groups of surfacing African penguins and have been observed feeding on small fish in these situations [8]. We investigated the potential mechanisms of facilitation between African penguins and volant seabirds from an *in situ* perspective by analysing footage of animal-borne video recorders (AVRs) deployed on breeding African penguins at Stony Point, South Africa. Specifically, we tested the hypothesis that interspecific encounters are driven by volant seabirds actively seeking out deep-diving birds to access their ability to herd prey to the surface.

## 2. Methods

African penguins at Stony Point, South Africa, were fitted with AVRs during the guard phase of four breeding seasons (June to August) between 2015 and 2018. Details of the AVR specifications and programme schedules are provided in electronic supplementary material, appendix S1. During 2017 and 2018 birds were additionally fitted with Axy Depth loggers (ADL, TechnoSmart, Rome, dimensions: length × width × height, 35 × 14 × 10 mm, weight: 6.5 g). Devices were attached to the lower backs of African penguins with Tesa tape (Beiersdorf AG, Hamburg, Germany) during the late afternoon preceding a foraging trip and were removed once the birds had returned to their nests. Previous use of AVRs on African penguins showed no adverse effects [7]. All observations recorded from the AVRs were limited by the optical performance of these devices: an angle of view of *ca* 110° and detection of seabirds up to 50 m at the surface (electronic supplementary material, appendix S1).

The raw footage was analysed in VLC media player (VideoLAN, France) and classified into dive and surface events. Dive bouts were classified as a sequence of more than four dives greater than 3 m deep with inter-dive durations less than 75 s, i.e. the bout ending criteria (BEC). Calculation of the BEC followed [9] using maximum-likelihood estimation criteria calculated from the dive parameters using R [10] package 'diveMove' [11]. BECs were calculated for all birds equipped with ADLs in 2017 and the maximum BEC value, i.e. 75 s, was used as the threshold for defining dive bouts for all birds. Dive depths for birds not equipped with ADLs were estimated using a descent rate of $1.22 \text{ m s}^{-1}$ [12]. Interspecific encounters were recorded for each dive and surface event including counts and identification to the lowest taxonomic level possible, encounter mode (flight, surface and prey pursuit/catch) and distance proximity (close—*ca* less than 15 m, far—more than 15 m, see electronic supplementary material, appendix S1 and figure S1 for rationale). For each event the number of fish caught by the penguin and the number of conspecifics were recorded.

Based on the hypothesis that volant seabirds cue on diving seabirds we predicted that the time elapsed to the first interspecific encounter (close encounters less than 15 m) in a dive bout would be inversely proportional to penguin group size with greater numbers being associated with increased visibility. We used linear mixed effects models (LMM) to test for significant differences between time elapsed to first encounters and two measures of penguin group size (to account for group sizes changing during a dive bout): (i) the maximum number of penguins recorded during a dive bout (max. group size) and (ii) the maximum number of penguins seen in the first 5 min of a dive bout (initial group size). Encounter mode, i.e. flight versus surface, was included as a fixed effect and the total number of fish caught by a penguin in a dive bout was included as a covariate to control for potential variation in productivity and its influence on the response (see below). For all models, bird ID was included as a random effect to account for potential pseudoreplication between observations from the same individual. All responses were log transformed and the LMMs were fitted using R package 'lme4' [13].

In the above context, the frequency of interspecific associations could potentially be independent of volant birds cuing on penguins, but rather be a function of other factors related to prey abundance, such as olfactory cues driving the high incidence of procellarids [14]. Assuming that catch rates of African penguins reflect relative prey abundance, *sensu* for little penguins *Eudyptula minor* see [15], we tested

**Table 1.** Species and groups recorded by animal-borne video recorders on African penguins.

| nomenclature | | | | group size | |
| --- | --- | --- | --- | --- | --- |
| scientific | common | group | obs. (N) | mean | s.d. |
| Procellariidae sp. | petrel, shearwater | Procellariidae | 128 | 2.5 | 2.0 |
| Puffinus griseus | sooty shearwater | Procellariidae | 103 | 3.8 | 3.8 |
| Diomedeidae sp. | albatross | n.a. | 1 | — | — |
| Morus capensis | Cape gannet | n.a. | 1 | — | — |
| Phalacrocorax capensis | Cape cormorant | Cape cormorants | 156 | 4.3 | 4.2 |
| Larus sp. | gull | n.a. | 2 | 3.5 | 2.1 |
| Larus vetula | kelp gull | n.a. | 6 | 2.3 | 1.8 |
| Sterninae sp. | tern | Terns | 69 | 7.7 | 6.1 |
| Thalasseus bergii | swift tern | Terns | 34 | 10.6 | 10.6 |

the potential for this confounding influence by conducting Spearman's rank correlation tests between the total number of prey caught by penguins during a dive bout (a proxy for relative prey abundance) and two measures of interspecific encounter rates: maximum number of volant seabirds recorded and a residency index (RI) for each species group for all observations, close and far. RI was calculated as the proportion of surface events in a dive bout that included interspecific encounters. Only complete dive bouts, i.e. separated by the BEC at start and end, were used in the analyses.

The prey pursuit sequences of African penguins were interpreted for all catch events involving other seabirds. Prey pursuit sequences were classified based on the location and transition of schooling fish during penguin pursuits: 'elevated schools'—schools located at depth and herded to shallower waters [7], and 'shallow schools'—schools initially encountered at or near the surface (less than 5 m). For each event, the difference in time between the first instance of prey location by penguins and the first observed prey pursuit by other seabirds was calculated to investigate the occurrence of facilitation.

## 3. Results

Video footage used in the analyses included 31 h of recordings from 20 individuals (mean: 93, range: 26–536 min per individual), including 57 complete dive bouts (mean: 3, range: 1–21 dive bouts per individual) from 19 individuals. Interspecific interactions included 500 sightings from nine taxonomic groups (table 1). Due to sample size restrictions, only three groups were used in the subsequent analyses: (i) Procellariidae, (ii) Cape cormorants *Phalacrocorax capensis* and (iii) terns (table 1). A large proportion (83%) of penguin dive bouts included interspecific encounters of which birds in flight (as opposed to on the water surface) were the most frequently observed (51%), especially Procellariidae, which were significantly more prevalent than other seabird groups (Chi-squared test, $\chi^2 = 8.4$, $p = 0.02$, electronic supplementary material, figure S2). Cape cormorants were recorded more frequently on the water surface than in flight, and for all seabird groups, catch events involving volant seabirds constituted the smallest proportion (10%) of dive bout interactions (electronic supplementary material, figure S2). African penguin group size was significantly inversely related to the time elapsed from the onset of dive bouts to first encounters with volant seabirds; this relationship held for models using both estimates of penguin group size (table 2, figure 1). Encounter mode had a weak influence on this response but the total number of fish caught by penguins in a dive bout had a significant positive influence with seabirds being encountered later in dive bouts during more productive periods (table 2, figure 1).We found no significant associations between volant seabird and penguin encounter rates (max. number and RI) and estimates of fish abundance (table 3). However we did find significant positive correlations between tern and Procellariidae numbers and the maximum number of African penguins recorded for each dive bout (table 3).

Active pursuit of prey by other seabirds was recorded in 10 events, all involving fish schools (electronic supplementary material, table S1). The majority (70%) of these prey pursuit sequences involved elevated schools with prey initially being detected by penguins at depths greater than 33 m, and subsequently pursued by volant seabirds 17 to 154 s later, by which time schools had been

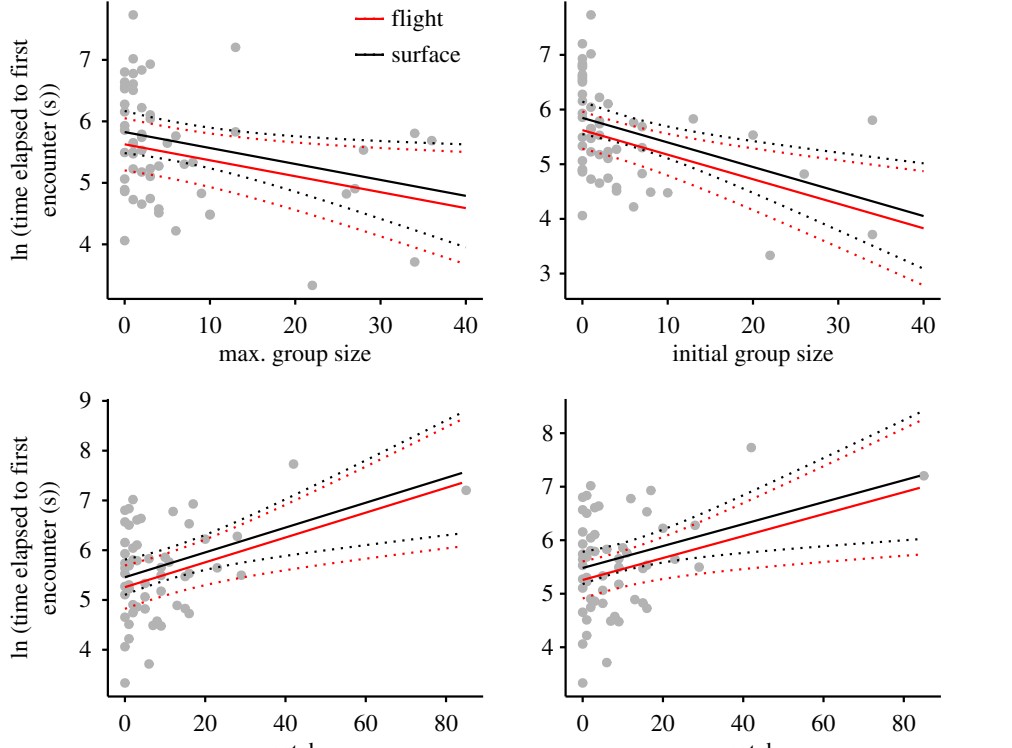

**Figure 1.** Influence of penguin group size (top panel) and total number of fish caught (bottom panel) by African penguins on the time elapsed since the onset of a dive bout to the first encounter (flight and at the surface) with volant seabird(s). Tests of both responses are modelled against the maximum group size (max. group size) of penguins observed during a dive bout and the maximum number observed during the first 5 min of a dive bout (initial group size). Fitted regressions estimated from linear mixed effects models are shown for first encounters with birds in flight and at the surface; dotted lines represent 95% confidence intervals.

**Table 2.** Linear mixed effects model predictions for the influence of African penguin group size (maximum group size [max. group size] of penguins observed during a dive bout and the maximum number observed during the first 5 min of a dive bout [initial group size]) on the time elapsed to first encounters with volant seabirds. Encounter mode (surface versus flight) and total number of fish caught in a dive bout are included as explanatory variables. Coefficients ($\beta$), standard errors (s.e.), $t$-statistics and $p$-values (significant values at 5% in italics) are given.

| group size variable | explanatory | $\beta$ | s.e. | $t$ | $p$ |
|---|---|---|---|---|---|
| max. group size | group size | −0.03 | 0.01 | −2.24 | *0.03* |
| max. group size | mode (surface) | 0.2 | 0.23 | 0.88 | 0.38 |
| max. group size | catch | 0.03 | 0.01 | 3.08 | *0.003* |
| initial group size | group size | −0.05 | 0.01 | −3.3 | *0.002* |
| initial group size | mode (surface) | 0.22 | 0.22 | 1.03 | 0.31 |
| initial group size | catch | 0.02 | 0.01 | 2.59 | *0.01* |

herded by penguins to within *ca* 5 m of the surface (electronic supplementary material, table S1; movie S1). Procellariidae and Cape cormorants were the most frequent participants in prey pursuit events ($n = 8$, 7 respectively), while terns were only observed in four events. Footage from bird SP1801 (8.9 h of recordings) included five interspecific prey pursuit sequences: four elevated and one shallow school event (figure 2a; electronic supplementary material, table S1). Catches by this penguin from schools were taken at significantly shallower depths than single fish prey (depth, median $\pm$ interquartile range: schools—15 $\pm$ 19, $n = 104$; single—30 $\pm$ 35, $n = 80$, Wilcoxon rank sum test, $W = 2693$, $p < 0.001$). Surface encounters with volant seabirds during elevated school events only occurred after the schools were driven to the surface by penguins. Prey pursuits by volant seabirds were only observed in subsequent shallow dives up to 2.6 min after the school was initially located by the penguin at depths greater than 33 m (figure 2b; electronic supplementary material, table S1; movie S1).

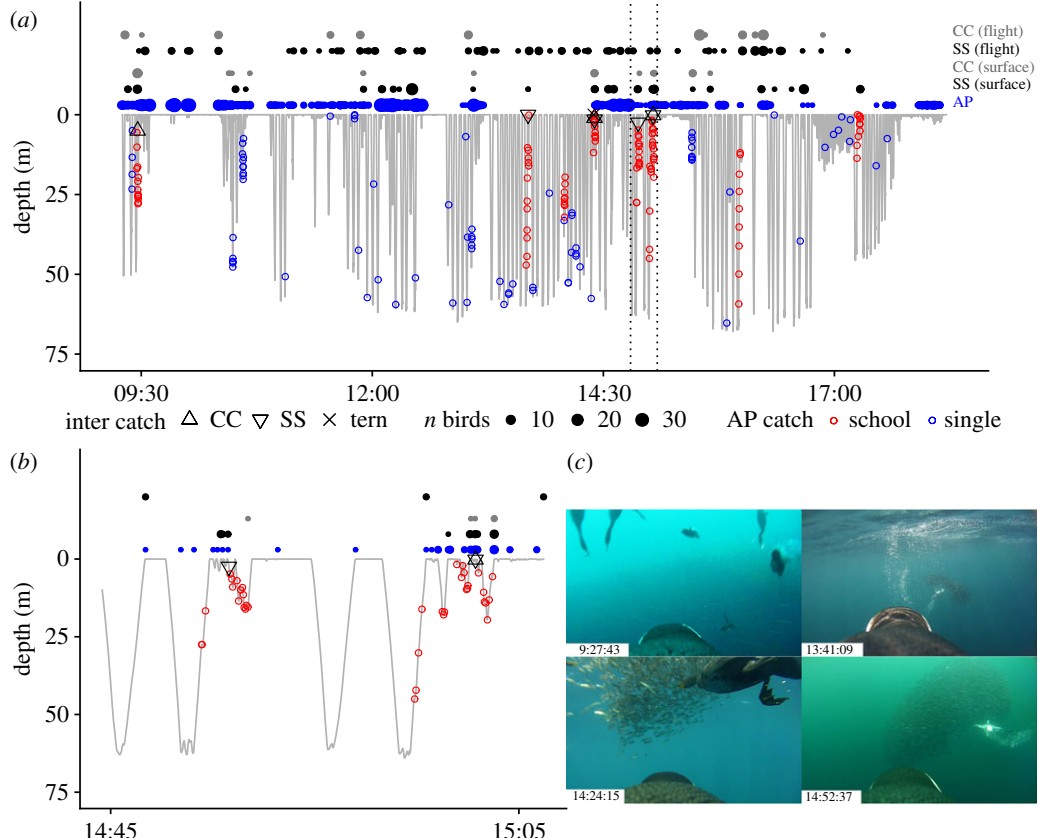

**Figure 2.** Dive chronology of penguin SP1801: (*a*) entire foraging trip recorded, (*b*) enlarged timeslot demarcated as vertical dotted lines in (*a*), and (*c*) images from AVRs indexed to time. The number of birds (*n* birds) in flight and at the surface are shown: Cape cormorants (CC, grey), sooty shearwaters (SS, black) and African penguins (AP, blue). Prey captures by volant seabirds (inter catch, tern = swift tern) and by African penguins (AP catch for schooling and single fish) are superimposed on the dive profile. Images show CC (left panel) and SS (right panel) prey pursuits/catches: sooty shearwaters are located at terminal ends of bubble trails foraging at the periphery of the fish schools. Confirmation of species identification of volant seabirds confirmed from images in sequence not displayed.

**Table 3.** Spearman's rank correlation tests between proxies for interspecific encounters (maximum number of volant seabirds recorded and residency index) and two explanatory variables: total number of prey caught by African penguins in a dive bout (AP catch) and maximum number of African penguins recorded in a dive bout (max. AP). Correlation coefficients ($r_s$) and *p*-values (significant values at 5%, in italics) are given.

| | response | | | explanatory | | | |
| | | | | AP catch | | max. AP | |
| group | variable | median (range) | N dive bouts | $r_s$ | p | $r_s$ | p |
|---|---|---|---|---|---|---|---|
| Procellariidae | max. number | 3 (1−25) | 31 | −0.002 | 0.9 | 0.35 | 0.05 |
| | residency index | 0.3 (0−0.75) | | −0.2 | 0.4 | 0.19 | 0.3 |
| Cape cormorants | max. number | 4 (1−20) | 26 | −0.22 | 0.3 | 0.19 | 0.4 |
| | residency index | 0.2 (0−0.8) | | −0.3 | 0.1 | 0.16 | 0.5 |
| Terns | max. number | 6 (1−40) | 15 | 0.04 | 0.9 | 0.63 | *0.01* |
| | residency index | 0.3 (0.05−1) | | 0.08 | 0.8 | 0.13 | 0.6 |

## 4. Discussion

Schooling fish constitute the most profitable prey to African penguins where they are often herded to shallower depths and accessed more repeatedly in multiple shallow dives ultimately improving foraging efficiency [7]. Results of this study have shown that these elevated school events also benefit

other seabirds, such as Cape cormorants, sooty shearwaters (*Puffinus griseus*) and swift terns (*Thalasseus bergii*) through facilitation. Frequent close (less than 15 m) surface and flight encounters not involving prey pursuits suggest that these species may be actively showing interest in penguins at the surface and are potentially getting a closer view on availability of prey. For the only bird that we had an almost complete foraging trip recorded (figure 2), it is significant to note that 63% of this bird's schooling prey pursuits involved other seabirds targeting the same prey.

The disproportionately high number of flight encounters with Procellariidae species observed in this study may have been influenced by higher densities of these species in the region during winter [16]. Detection probabilities of the penguin AVRs are also likely to be biased to the arcing flight patterns of procellarids in comparison to the more directional flights of Cape cormorants. This may explain the discrepancy in frequencies of flight and surface encounters between these two groups (electronic supplementary material, figure S2).

The findings of this study provide evidence that penguins facilitate prey capture by volant seabirds. This form of association may play an important role in the foraging efficiency of volant seabirds whose distributions overlap those of facilitating divers [5]. The extent to which diving species contribute to overall foraging success of volant species will depend on the overlap in their foraging ranges, dietary specialization and diving capabilities. However, without an empirical understanding of prey capture rates and energy expenditure by volant seabird species that benefit from facilitation, the relative contribution that diving seabirds provide to the foraging success of these species remains speculative.

During this study, interspecific encounters between penguins and volant seabirds were more frequent when proxies for prey availability and abundance were relatively low and these encounters also occurred sooner during a penguin dive bout under these conditions (table 2). This may reflect the influence of regional prey abundance on the distribution of seabirds with higher densities of birds closely tracking fewer patches of prey when prey availability is low (e.g. [17]). In such conditions when volant seabirds' foraging ranges overlap those of breeding penguins it may be advantageous to actively seek out surfacing penguins who are known to track the distribution of their prey effectively around their colonies [15]. For Cape cormorants that breed alongside African penguins at Stony Point during September and October this may be especially profitable. When attending young chicks, Cape cormorants undergo frequent short foraging trips within foraging ranges very similar to those of breeding African penguins [18]. Under the constraints of central place foraging (*sensu* [19]) and relatively high energetic costs associated with high wing loading [20], to meet the energetic demands of growing chicks, Cape cormorants may benefit from tracking the locations of foraging African penguins when prey patches are difficult to find. Given the similarities in the distributions and diet of African penguins and Cape cormorants [21], facilitation by the former species may play a significant role in the foraging efficiency of the latter.

While volant seabirds may benefit from diving species, it is not clear if this relationship incurs any disruption or benefit to the foraging abilities of the diving species. For diving species that benefit from group foraging, such as African penguins, disruption of school cohesiveness near the surface, maintained by the corralling behaviour of conspecifics [7], may incur a net reduction in prey capture rates if fish are prematurely released from their hold. A bolstered sample of interspecific catch events may clarify the relative influence these interactions have on the foraging efficiency of facilitating divers. Nonetheless, it is clear that penguins can benefit seabird communities, and their changing population numbers potentially have far-reaching implications in marine ecosystems.

Ethics. All fieldwork carried out on African penguins at Stony Point was done under permission from the South African Department of Environmental Affairs (permit nos. RES 2015/38, RES 2016/100, RES 2017/21, RES 2018/36) and Cape Nature (permit nos. AAA007-00209-0056, 0056-AAA007-00218). All procedures were approved by the Nelson Mandela Metropolitan University's Animal Ethics Committee (Ethics clearance reference no. A17-SCI-ZOO-001).

Data accessibility. Data used in the analyses can be accessed from the Dryad Digital Repository: https://doi.org/10.5061/dryad.5q04b32 [22]. Electronic supplementary material, movie S1 can be accessed at: https://drive.google.com/open?id=1-LphMLLH4vc_JjrqTQMEBXuWEDijD43y.

Authors' contributions. Both authors conceived of and designed the study. A.M.M. coordinated the fieldwork, conducted the statistical analyses and drafted the manuscript. Both authors revised and gave final approval of the manuscript.

Competing interests. The authors declare no competing interests.

Funding. Funding for this research was made possible by Nelson Mandela University and the DST/NRF Centre of Excellence at the Percy FitzPatrick Institute. Funding for AVRs was provided by the Saving Animals from Extinction (SAFE) project of the Association of Zoos and Aquariums (AZA).

Acknowledgements. We would like to thank Peter Ryan for identification of seabird species from the footage and for useful insights into the behaviour of seabirds at sea. We thank the staff of Cape Nature, especially Cuan McGeorge, for their help in the field. Three anonymous reviewers are thanked for their valuable comments on the original manuscript.

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
