## [Reviewer comments · Royal Society Open Science]

Review History

RSOS-190333.R0 (Original submission)

Review form: Reviewer 1

Is the manuscript scientifically sound in its present form?

Yes

Are the interpretations and conclusions justified by the results?

No

Is the language acceptable?

Yes

Is it clear how to access all supporting data?

Yes

Do you have any ethical concerns with this paper?

No

Have you any concerns about statistical analyses in this paper?

I do not feel qualified to assess the statistics

Recommendation?

Accept with minor revision (please list in comments)

Comments to the Author(s)

Dear authors,

Thank you for this good revision of your fascinating paper. I have a few small suggestions for your consideration.

At line 31 (penultimate of first paragraph in Introduction use 'the study has until'.

Lines 54 and 55 (third and fourth of second paragraph of Methods) have awkward construction: 'estimation criteria estimated from the dive parameters quantified using'. Would 'criteria estimated from the dive parameters using' say the same thing?

At line 66, and in the caption to Fig. 1, it may be worth mentioning that group size refers to penguins additional to the one with the AVR (at least I assume so). (Apologies that I did not mention this earlier.)

At lines 67 and 69 (fourth and second last of third paragraph of Methods) add '!' after 'estimates' and 'relationship'.

Line 95 (last of first paragraph of Results) - should this be 'involving seabirds other than penguins'?

Line 102 (second last of second paragraph of Results). Should this not read 'were significantly positively related'?

Line 109 (4 from end of p 4) could be shortened: 'SP1801 (8.9 h of recordings) included'.

Third last line on p 4 - Is this 'catches by penguins'; if so or not it should be made clear.

Second last line on p 4 - please advise what 'IQR' stands for (Again, apologies that I did not mention this earlier.)

Line 133 (second of third paragraph in Discussion) has an errant apostrophe after 'seabirds'.

At lines 140 and 141 (at start of last paragraph), is it not possible that volant birds once in the water could also assist in keeping a shoal corralled and near the surface? I.e. there could be cost or benefit to penguins.

In Fig 2B is there any x for tern? (I could not spot one.) If not, this could be removed from inter catch and the caption could be adjusted accordingly.

Thank you.

Review form: Reviewer 2**Is the manuscript scientifically sound in its present form?**

No

Are the interpretations and conclusions justified by the results?

Yes

Is the language acceptable?

Yes

Is it clear how to access all supporting data?

Yes

Do you have any ethical concerns with this paper?

No

Have you any concerns about statistical analyses in this paper?

Yes

Recommendation?

Major revision is needed (please make suggestions in comments)

Comments to the Author(s)

General comments

The manuscript N° RSOS-190333 presented by Drs. McInnes and Pistorius entitled: "Up for grabs - prey herding by penguins facilitates shallow foraging by volant seabirds", presents novel information obtained through footage from animal-borne video loggers mounted on a species of penguin categorized by the IUCN Red List of Threatened Species as endangered; with the intention of evaluating the foraging facilitation that these diving birds produce on other species of volant seabirds. Although this inter-specific relation is known by anyone who inhabits or uses pelagic marine regions (fishermen, sailors, divers, researchers, etc.), this is the first time that such behaviour is empirically tested, as the authors very well point out. The work presents a well formulated and parsimonious hypothesis, which was tested by contrasting its prediction against the empirically obtained data. This makes the work epistemologically correct and neat, despite not having a more in-depth discussion about the results found. However, and in relation to the latter, the statistical analysis used requires a thorough revision. Especially to clarify some issues that will be detailed in the specific comments. Finally, the most important result of this work is the verification (albeit partial) of the role of facilitators that penguins that feed on schools have respect to flying seabirds (and shallow divers). This is a result that highlights again the central role that these fabulous seabirds have in the structuring and functioning of the southern marine ecosystems. But, according to me, there are several issues to be considered, which were detailed in the "specific comments" section, before this manuscript could be completely suitable for publication. Please see the comments below.

Specific comments

INTRODUCTION

P2, L-WHO KNOWS (WK): "...waters [2,4].The implications", missing a space after the point.

P2, L-WK: "...upwelling ecosystem where they feed", there is an extra space before "where".

P2, L-WK: "...by analysing footage of Animal-borne video recorders (AVR)", change "Animal" by "animal".

METHODS

Here is an important aspect to be revisited. The authors have to explain better the use of the statistics they did. Why do they use nonparametric statistics (chi-square and Wilcoxon) to evaluate repeated samples of particular individuals (i.e. pseudo-replication)? The authors will understand that each bout, or independent dive, belonging to a particular individual is a pseudo-replicate unless they use a single bout (or dive) per ID, and that is not what they did.

P3, L-WK: "We used the Kaplan-Meier estimate in R package 'survival' [13] to test for...". Why did they use a package linked to survival to analyse the times of the first interspecific encounter? I recommend that the authors introduce at least one line to clarify it. In addition, they must consider if that test is possible to be used taking into account the comment above. Also, what are

the assumptions of this test? Do your data comply with these assumptions? If not, what did you do to save this problem?

RESULTS

P4, L-WK: "...including 57 complete dive bouts (mean \pm SD: 3 ± 5 per individual) from 19 individuals." According to this sentence, it is seen that the authors group the bouts of the 19 IDs all together (N = 57) to perform the analyses. Would not they be incurring in pseudo-replication?

P4, L96?-102?: From "African penguin group size... (to) ... for each dive bout (Table 2)." Could the authors see to rewrite these results through a biological/ecological interpretation of them and not in statistical terms? The methodology used to contrast the prediction of their hypothesis is somewhat complex so that the reader, not seasoned in these issues, can deduce from their results the support or not to the hypothesis they are testing.

P4, L104?-107?: Exactly like this you should express the results of the previous paragraph. Excellent.

P4, L106?: "depths > 33 m", but where does the 33 m cutting line come from? The authors do not say where that number comes from or I cannot find it... Is it an average, a mode, or a median? And from what?

P4, L111?: Please, define what IQR is, it is not anywhere.

DISCUSSION

In my opinion, the discussion deserves to be improved. As it is, it is merely an extension of the results (which are rather brief...) and is not discussed with the bibliography, not even with the one cited in the introduction.

P5, L132?-135?: The symbiotic association implies a mutual benefit. This it does not seem to be the case, since penguins would have no benefit from the actions of flying seabird species (at least the authors do not explain it at work). Without going any further, we have (video) records of kleptoparasitism from gulls to penguins in these foraging flocks events (unpublished data).

P5, L138?-139?: "...in the foraging efficiency of the latter." I would add something like this: "and an expected decrease in the energy expenditure associated to flight and diving, an onerous investment in this group of seabirds (REFs, there is a bunch)".

P5, L140?-141?: Then you can not talk about symbiosis, in any way. It is just a positive association or facilitation seen from the foraging ecology of flying seabirds.

P5, L142?: There is an extra space after "penguins" and before the comma.

FIGURES

Figure 1 - P8, L213?-214?: Authors should explain this in the methods (also).

Decision letter (RSOS-190333.R0)

01-Apr-2019

Dear Dr McInnes,

The editors assigned to your paper ("Up for grabs - prey herding by penguins facilitates shallow foraging by volant seabirds") have now received comments from reviewers. We would like you to revise your paper in accordance with the referee and Associate Editor suggestions which can be found below (not including confidential reports to the Editor). Please note this decision does not guarantee eventual acceptance.

Please submit a copy of your revised paper before 24-Apr-2019. Please note that the revision deadline will expire at 00.00am on this date. If we do not hear from you within this time then it will be assumed that the paper has been withdrawn. In exceptional circumstances, extensions may be possible if agreed with the Editorial Office in advance. We do not allow multiple rounds of revision so we urge you to make every effort to fully address all of the comments at this stage. If deemed necessary by the Editors, your manuscript will be sent back to one or more of the original reviewers for assessment. If the original reviewers are not available, we may invite new reviewers.

- Data accessibility

If you wish to submit your supporting data or code to Dryad (<http://datadryad.org/>), or modify your current submission to dryad, please use the following link:
<http://datadryad.org/submit?journalID=RSOS&manu=RSOS-190333>

- **Competing interests**

- **Authors' contributions**

- **Acknowledgements**

- **Funding statement**

on behalf of Professor Kevin Padian (Subject Editor)
openscience@royalsociety.org

Comments to Author:

Reviewers' Comments to Author:

Reviewer: 1

Comments to the Author(s)

Dear authors,

Thank you for this good revision of your fascinating paper. I have a few small suggestions for your consideration.

At line 31 (penultimate of first paragraph in Introduction use 'the study has until').

Lines 54 and 55 (third and fourth of second paragraph of Methods) have awkward construction: 'estimation criteria estimated from the dive parameters quantified using'. Would 'criteria estimated from the dive parameters using' say the same thing?

At line 66, and in the caption to Fig. 1, it may be worth mentioning that group size refers to penguins additional to the one with the AVR (at least I assume so). (Apologies that I did not mention this earlier.)

At lines 67 and 69 (fourth and second last of third paragraph of Methods) add ':' after 'estimates' and 'relationship'.

Line 95 (last of first paragraph of Results) - should this be 'involving seabirds other than penguins'?

Line 102 (second last of second paragraph of Results). Should this not read 'were significantly positively related'?

Line 109 (4 from end of p 4) could be shortened: 'SP1801 (8.9 h of recordings) included'.

Third last line on p 4 - Is this 'catches by penguins'; if so or not it should be made clear.

Second last line on p 4 - please advise what 'IQR' stands for (Again, apologies that I did not mention this earlier.)

Line 133 (second of third paragraph in Discussion) has an errant apostrophe after 'seabirds'.

At lines 140 and 141 (at start of last paragraph), is it not possible that volant birds once in the water could also assist in keeping a shoal corralled and near the surface? I.e. there could be cost or benefit to penguins.

In Fig 2B is there any x for tern? (I could not spot one.) If not, this could be removed from inter catch and the caption could be adjusted accordingly.

Thank you.

Reviewer: 2

Comments to the Author(s)
General comments

The manuscript N° RSOS-190333 presented by Drs. McInnes and Pistorius entitled: “Up for grabs - prey herding by penguins facilitates shallow foraging by volant seabirds”, presents novel information obtained through footage from animal-borne video loggers mounted on a species of penguin categorized by the IUCN Red List of Threatened Species as endangered; with the intention of evaluating the foraging facilitation that these diving birds produce on other species of volant seabirds. Although this inter-specific relation is known by anyone who inhabits or uses pelagic marine regions (fishermen, sailors, divers, researchers, etc.), this is the first time that such behaviour is empirically tested, as the authors very well point out. The work presents a well formulated and parsimonious hypothesis, which was tested by contrasting its prediction against the empirically obtained data. This makes the work epistemologically correct and neat, despite not having a more in-depth discussion about the results found. However, and in relation to the latter, the statistical analysis used requires a thorough revision. Especially to clarify some issues that will be detailed in the specific comments. Finally, the most important result of this work is the verification (albeit partial) of the role of facilitators that penguins that feed on schools have respect to flying seabirds (and shallow divers). This is a result that highlights again the central role that these fabulous seabirds have in the structuring and functioning of the southern marine ecosystems. But, according to me, there are several issues to be considered, which were detailed in the “specific comments” section, before this manuscript could be completely suitable for publication. Please see the comments below.

Specific comments

INTRODUCTION

P2, L-WHO KNOWS (WK): "...waters [2,4].The implications", missing a space after the point.

P2, L-WK: "...upwelling ecosystem where they feed", there is an extra space before "where".

P2, L-WK: "...by analysing footage of Animal-borne video recorders (AVR)", change "Animal" by "animal".

METHODS

Here is an important aspect to be revisited. The authors have to explain better the use of the statistics they did. Why do they use nonparametric statistics (chi-square and Wilcoxon) to evaluate repeated samples of particular individuals (i.e. pseudo-replication)? The authors will understand that each bout, or independent dive, belonging to a particular individual is a pseudo-replicate unless they use a single bout (or dive) per ID, and that is not what they did.

P3, L-WK: "We used the Kaplan-Meier estimate in R package 'survival' [13] to test for...". Why did they use a package linked to survival to analyse the times of the first interspecific encounter? I recommend that the authors introduce at least one line to clarify it. In addition, they must consider if that test is possible to be used taking into account the comment above. Also, what are the assumptions of this test? Do your data comply with these assumptions? If not, what did you do to save this problem?

RESULTS

P4, L-WK: "...including 57 complete dive bouts (mean \pm SD: 3 ± 5 per individual) from 19 individuals." According to this sentence, it is seen that the authors group the bouts of the 19 IDs all together ($N = 57$) to perform the analyses. Would not they be incurring in pseudo-replication?

P4, L96?-102?: From "African penguin group size... (to) ... for each dive bout (Table 2)." Could the authors see to rewrite these results through a biological/ecological interpretation of them and not in statistical terms? The methodology used to contrast the prediction of their hypothesis is somewhat complex so that the reader, not seasoned in these issues, can deduce from their results the support or not to the hypothesis they are testing.

P4, L104?-107?: Exactly like this you should express the results of the previous paragraph. Excellent.

P4, L106?: "depths > 33 m", but where does the 33 m cutting line come from? The authors do not say where that number comes from or I cannot find it... Is it an average, a mode, or a median? And from what?

P4, L111?: Please, define what IQR is, it is not anywhere.

DISCUSSION

In my opinion, the discussion deserves to be improved. As it is, it is merely an extension of the results (which are rather brief...) and is not discussed with the bibliography, not even with the one cited in the introduction.

P5, L132?-135?: The symbiotic association implies a mutual benefit. This it does not seem to be the

case, since penguins would have no benefit from the actions of flying seabird species (at least the authors do not explain it at work). Without going any further, we have (video) records of kleptoparasitism from gulls to penguins in these foraging flocks events (unpublished data).

P5, L138?-139?: "...in the foraging efficiency of the latter." I would add something like this: "and an expected decrease in the energy expenditure associated to flight and diving, an onerous investment in this group of seabirds (REFs, there is a bunch)".

P5, L140?-141?: Then you can not talk about symbiosis, in any way. It is just a positive association or facilitation seen from the foraging ecology of flying seabirds.

P5, L142?: There is an extra space after "penguins" and before the comma.

FIGURES

Figure 1 - P8, L213?-214?: Authors should explain this in the methods (also).

Author's Response to Decision Letter for (RSOS-190333.R0)

See Appendix A.

RSOS-190333.R1 (Revision)

Review form: Reviewer 2

Is the manuscript scientifically sound in its present form?

Yes

Are the interpretations and conclusions justified by the results?

Yes

Is the language acceptable?

Yes

Is it clear how to access all supporting data?

Yes

Do you have any ethical concerns with this paper?

No

Have you any concerns about statistical analyses in this paper?

No

Recommendation?

Accept as is

Comments to the Author(s)

Dear Authors:

Your revised manuscript N^o RSOS-190333.R1 entitled: "Up for grabs - prey herding by penguins facilitates shallow foraging by volant seabirds", presents substantial improvements. You have taken the hard work of modifying the statistical approach (with the subsequent changes in the results section) and expanded the discussion. I really believe that now the manuscript has improved a lot and is ready to be published in this prestigious journal.

Best wishes and congratulations on this brilliant work,

Decision letter (RSOS-190333.R1)

14-May-2019

Dear Dr McInnes,

I am pleased to inform you that your manuscript entitled "Up for grabs - prey herding by penguins facilitates shallow foraging by volant seabirds" is now accepted for publication in Royal Society Open Science.

on behalf of Prof Kevin Padian (Subject Editor)
openscience@royalsociety.org

Reviewer comments to Author:

Reviewer: 2

Comments to the Author(s)

Dear Authors:

Your revised manuscript N^o RSOS-190333.R1 entitled: "Up for grabs - prey herding by penguins facilitates shallow foraging by volant seabirds", presents substantial improvements. You have taken the hard work of modifying the statistical approach (with the subsequent changes in the results section) and expanded the discussion. I really believe that now the manuscript has improved a lot and is ready to be published in this prestigious journal.

Best wishes and congratulations on this brilliant work,

Appendix A

Reviewers' report

Up for grabs - prey herding by penguins facilitates shallow foraging by volant seabirds

McInnes, AM and Pistorius, PA

The following contains responses to the referees' concerns regarding the manuscript. The authors' responses are highlighted in bold type for each comment. Line references in the responses refer to the revised manuscript.

Reviewer 1:

At line 31 (penultimate of first paragraph in Introduction use 'the study has until'.

This has been changed as recommended.

Lines 54 and 55 (third and fourth of second paragraph of Methods) have awkward construction: 'estimation criteria estimated from the dive parameters quantified using'. Would 'criteria estimated from the dive parameters using' say the same thing?

This sentence has been changed to:

Lines 54 - 55: Calculation of the BEC followed [9] using maximum likelihood estimation criteria calculated from the dive parameters using R [10] package 'diveMove' [11].

At line 66, and in the caption to Fig. 1, it may be worth mentioning that group size refers to penguins additional to the one with the AVR (at least I assume so). (Apologies that I did not mention this earlier.)

This is no longer relevant as the statistical approach was revised in accordance with reviewer 2's concerns.

At lines 67 and 69 (fourth and second last of third paragraph of Methods) add ':' after 'estimates' and 'relationship'.

These have been added as suggested.

Line 95 (last of first paragraph of Results) - should this be 'involving seabirds other than penguins'?

This has been changed to:

Lines 95 - 98: Cape cormorants were recorded more frequently on the water surface than in flight, and for all seabird groups, catch events involving volant seabirds constituted the smallest proportion (10 %) of dive bout interactions (Figure S2).

Line 102 (second last of second paragraph of Results). Should this not read 'were significantly positively related'?

This section has been changed to:

Lines 102 - 105: We found no significant associations between seabird encounter rates (max number and RI) and estimates of fish abundance (Table 3). However we did find significant positive correlations between tern and Procellariidae numbers and the maximum number of African penguins recorded for each dive bout (Table 3).

Line 109 (4 from end of p 4) could be shortened: 'SP1801 (8.9 h of recordings) included'.

This has been changed to:

Lines 110 - 111: Footage from bird SP1801 (8.9 h of recordings) included five interspecific prey pursuit sequences: ...

Third last line on p 4 - Is this 'catches by penguins'; if so or not it should be made clear.

This has been clarified:

Line 112: Catches by this penguin from schools were...

Second last line on p 4 - please advise what 'IQR' stands for (Again, apologies that I did not mention this earlier.)

This has been written in full: interquartile range.

Line 133 (second of third paragraph in Discussion) has an errant apostrophe after 'seabirds'.

This has been removed as noted.

At lines 140 and 141 (at start of last paragraph), is it not possible that volant birds once in the water could also assist in keeping a shoal corralled and near the surface? I.e. there could be cost or benefit to penguins.

We have changed this sentence to:

Lines 152 - 153: While volant seabirds may benefit from pursuit diving species it is not clear if this relationship incurs any disruption or benefit to the foraging abilities of the facilitating species.

In Fig 2B is there any x for tern? (I could not spot one.) If not, this could be removed from inter catch and the caption could be adjusted accordingly.

The tern observation is at approx. 14:20 at the same time Cape cormorants and sooty shearwaters were observed - this is clearer in the pdf version.

Reviewer: 2

Comments to the Author(s)

General comments

The manuscript N^o RSOS-190333 presented by Drs. McInnes and Pistorius entitled: "Up for grabs - prey herding by penguins facilitates shallow foraging by volant seabirds", presents novel information obtained through footage from animal-borne video loggers mounted on a species of penguin categorized by the IUCN Red List of Threatened Species as endangered; with the intention of evaluating the foraging facilitation that these diving birds produce on other species of volant seabirds. Although this inter-specific relation is known by anyone who inhabits or uses pelagic marine regions (fishermen, sailors, divers, researchers, etc.), this is the first time that such behaviour is empirically tested, as the authors very well point out. The work presents a well formulated and parsimonious hypothesis, which was tested by contrasting its prediction against the empirically obtained data. This makes the work epistemologically correct and neat, despite not having a more in-depth discussion about the results found. However, and in relation to the latter, the statistical analysis used requires a thorough revision. Especially to clarify some issues that will be detailed in the specific comments. Finally, the most important result of this work is the verification (albeit partial) of the role of facilitators that penguins that feed on schools have respect to flying seabirds (and shallow divers). This is a result that highlights again the central role that these fabulous seabirds

have in the structuring and functioning of the southern marine ecosystems. But, according to me, there are several issues to be considered, which were detailed in the "specific comments" section, before this manuscript could be completely suitable for publication. Please see the comments below.

Specific comments

INTRODUCTION

P2, L-WHO KNOWS (WK): "...waters [2,4].The implications", missing a space after the point.

This has been inserted.

P2, L-WK: "...upwelling ecosystem where they feed", there is an extra space before "where".

This has been removed.

P2, L-WK: "...by analysing footage of Animal-borne video recorders (AVR)", change "Animal" by "animal".

This has been changed.

METHODS

Here is an important aspect to be revisited. The authors have to explain better the use of the statistics they did. Why do they use nonparametric statistics (chi-square and Wilcoxon) to evaluate repeated samples of particular individuals (i.e. pseudo-replication)? The authors will understand that each bout, or independent dive, belonging to a particular individual is a pseudo-replicate unless they use a single bout (or dive) per ID, and that is not what they did.

P3, L-WK: "We used the Kaplan-Meier estimate in R package 'survival' [13] to test for...". Why did they use a package linked to survival to analyse the times of the first interspecific encounter? I recommend that the authors introduce at least one line to clarify it. In addition, they must consider if that test is possible to be used taking into account the comment above. Also, what are the assumptions of this test? Do your data comply with these assumptions? If not, what did you do to save this problem?

We have revised our statistical approach in order to accommodate the valid concerns by the reviewer. We have replaced the Kaplan-Meier method with a linear mixed effects modelling approach to accommodate the potential violation of independence of samples.

We have added the following to the text:

Methods

Lines 64 - 72: . We used linear mixed effects models (LMM) to test for significant differences between time elapsed to first encounters and two measures of penguin group size (to account for

group sizes changing during a dive bout): (1) the maximum number of penguins recorded during a dive bout (max. group size) and (2) the maximum number of penguins seen in the first 5 min of a dive bout (initial group size). Encounter mode, i.e. flight versus surface, was included as a fixed effect and the total number of fish caught by a penguin in a dive bout was included as a covariate to control for potential variation in productivity and its influence on the response (see below). For all models bird ID was included as a random effect to account for potential pseudoreplication between observations from the same individual. All responses were log transformed and the LMMs were fitted using R package 'lme4' [13].

Results

Lines 98 - 102: African penguin group size was significantly inversely related to the time elapsed from the onset of dive bouts to first encounters with volant seabirds; this relationship held for models using both estimates of penguin group size (Table 1, Figure 1). Encounter mode had a weak influence on this response but the total number of fish caught by penguins in a dive bout had a significant positive influence with seabirds being encountered later in dive bouts during more productive periods (Table 1, Figure 1).

Figure 1. Influence of penguin group size (visibility) on the time elapsed since the onset of a dive bout to the first interspecific encounter. Tests of both responses are modelled against the maximum group size (max. group size) of penguins observed during a dive bout and the maximum number observed during the first 5 minutes of a dive bout (initial group size). Fitted regressions estimated from linear mixed effects models are shown for first encounters with birds in flight and at the surface; dotted lines represent 95 % confidence intervals.

Table 2. Linear mixed effects model predictions for the influence of African penguin group size (maximum recorded during dive bout and maximum recorded during initiation of dive bout) on

the time elapsed to first encounters with volant seabirds . Encounter mode (surface vs flight) and total number of fish caught in a dive bout are included as explanatory variables. Coefficients (β) , standard errors (SE), t-statistics and p-values (significant values at 5 % in bold) are given.

Group size variable	Explanatory	β	s.e.	t	p
max. group size	group size	-0.03	0.01	-2.24	0.03
max. group size	mode (surface)	0.2	0.23	0.88	0.38
max. group size	catch	0.03	0.01	3.08	0.003
initial group size	group size	-0.05	0.01	-3.3	0.002
initial group size	mode (surface)	0.22	0.22	1.03	0.31
initial group size	catch	0.02	0.01	2.59	0.01

RESULTS

P4, L-WK: "...including 57 complete dive bouts (mean \pm SD: 3 \pm 5 per individual) from 19 individuals." According to this sentence, it is seen that the authors group the bouts of the 19 IDs all together (N = 57) to perform the analyses. Would not they be incurring in pseudo-replication?

This has been addressed by changing the statistical approach (see above).

P4, L96?-102?: From "African penguin group size... (to) ... for each dive bout (Table 2)." Could the authors see to rewrite these results through a biological/ecological interpretation of them and not in statistical terms? The methodology used to contrast the prediction of their hypothesis is somewhat complex so that the reader, not seasoned in these issues, can deduce from their results the support or not to the hypothesis they are testing.

This has been changed to:

Lines 98- 105: African penguin group size was significantly inversely related to the time elapsed from the onset of dive bouts to first encounters with volant seabirds; this relationship held for models using both estimates of penguin group size (Table 1, Figure 1). Encounter mode had a weak influence on this response but the total number of fish caught by penguins in a dive bout had a significant positive influence with seabirds being encountered later in dive bouts during more productive periods (Table 1, Figure 1). We found no significant associations between seabird encounter rates (max number and RI) and estimates of fish abundance (Table 3). However we did find significant positive correlations between tern and Procellariidae numbers and the maximum number of African penguins recorded for each dive bout (Table 3).

P4, L104?-107?: Exactly like this you should express the results of the previous paragraph. Excellent.

Thanks!

P4, L106?: "depths > 33 m", but where does the 33 m cutting line come from? The authors do not say where that number comes from or I cannot find it... Is it an average, a mode, or a median? And from what?

This is deduced from Table S1 which is referenced at the end of the sentence.

P4, L111?: Please, define what IQR is, it is not anywhere.

This has been spelled out in full.

DISCUSSION

In my opinion, the discussion deserves to be improved. As it is, it is merely an extension of the results (which are rather brief...) and is not discussed with the bibliography, not even with the one cited in the introduction.

We have added a new paragraph to the discussion:

Lines 139 - 153: During this study interspecific encounters were more frequent when proxies for prey availability and abundance were relatively low and these encounters also occurred sooner during a dive bout under these conditions (Table 2). This may reflect the influence of regional prey abundance on the distribution of seabirds with higher densities of birds closely tracking fewer patches of prey when prey availability is low (e.g. [17]). In such conditions when volant seabirds' foraging ranges overlap those of breeding penguins it may be advantageous to actively seek out surfacing penguins who are known to track the distribution of their prey effectively around their colonies [15]. For Cape cormorants that breed alongside African penguins at Stony Point during September and October this may be especially profitable. When attending young chicks, Cape cormorants undergo frequent short foraging trips within a foraging range very similar to that of African penguins [18]. Under the constraints of central place foraging (sensu [19]) and relatively high energetic costs associated with high wing loading [20], to meet the energetic demands of growing chicks, Cape cormorants may benefit from tracking the locations of foraging African penguins when prey patches are difficult to find. Given the similarities in the distributions and diet of African penguins and Cape cormorants [21], facilitation by the former species may play a significant role in the foraging efficiency of the latter.

Including the following additional references:

17. Ainley DG, Dugger KD, Ford RG, Pierce SD, Reese DC, Brodeur RD, Tynan CT, Barth JA. 2009 Association of predators and prey at frontal features in the California Current: Competition, facilitation, and co-occurrence. *Mar. Ecol. Prog. Ser.* 389, 271–294.
18. Ryan PG, Pichegru L, Ropert-Coudert Y, Grémillet D, Kato A. 2010 On a wing and a prayer: The foraging ecology of breeding Cape cormorants. *J. Zool.* 280, 25–32.
19. Orians GH, Pearson NE. 1979 On the theory of central place foraging. In *Analysis of ecological systems* (eds DJ Horn, RD Mitchell, GR Stairs), pp. 154–177. Ohio State University Press, Columbus.
20. Enstipp MR, Jones DR, Lorentsen SH, Grémillet D. 2007 Energetic costs of diving and prey-capture capabilities in cormorants and shags (Phalacrocoracidae) underline their unique adaptation to the aquatic environment. *J. Ornithol.* 148, 593–600.

P5, L132?-135?: The symbiotic association implies a mutual benefit. This it does not seem to be the case, since penguins would have no benefit from the actions of flying seabird species (at least the authors do not explain it at work). Without going any further, we have (video) records of kleptoparasitism from gulls to penguins in these foraging flocks events (unpublished data).

We have replaced this term with 'association'.

P5, L138?-139?: "...in the foraging efficiency of the latter." I would add something like this: "and an expected decrease in the energy expenditure associated to flight and diving, an onerous investment in this group of seabirds (REFs, there is a bunch)".

We have added the following sentence which incorporates this and other aspects related to energetic costs:

Lines 147 - 150: Under the constraints of central place foraging (sensu [19]) and relatively high energetic costs associated with high wing loading [20], to meet the energetic demands of growing chicks, Cape cormorants may benefit from tracking the locations of foraging African penguins when prey patches are difficult to find.

P5, L140?-141?: Then you can not talk about symbiosis, in any way. It is just a positive association or facilitation seen from the foraging ecology of flying seabirds.

We have omitted any mention of symbiosis.

P5, L142?: There is an extra space after "penguins" and before the comma.

This has been removed.

FIGURES

Figure 1 - P8, L213?-214?: Authors should explain this in the methods (also).

This figure has been replaced and the caption updated accordingly.